# A Comprehensive Survey on Artifact Recovery from Social Media Platforms: Approaches and Future Research Directions

**Khushi Gupta, Damilola Oladimeji, Cihan Varol \*, Amar Rasheed and Narasimha Shahshidhar**

Department of Computer Science, Sam Houston State University, Huntsville, TX 77340, USA; kxg095@shsu.edu (K.G.); dko011@shsu.edu (D.O.); axr249@shsu.edu (A.R.); nks001@shsu.edu (N.S.)
\* Correspondence: cxv007@shsu.edu

**Abstract:** Social media applications have been ubiquitous in modern society, and their usage has grown exponentially over the years. With the widespread adoption of these platforms, social media has evolved into a significant origin of digital evidence in the domain of digital forensics. The increasing utilization of social media has caused an increase in the number of studies focusing on artifact (digital remnants of data) recovery from these platforms. As a result, we aim to present a comprehensive survey of the existing literature from the past 15 years on artifact recovery from social media applications in digital forensics. We analyze various approaches and techniques employed for artifact recovery, structuring our review on well-defined analysis focus categories, which are memory, disk, and network. By scrutinizing the available literature, we determine the trends and commonalities in existing research and further identify gaps in existing literature and areas of opportunity for future research in this field. The survey is expected to provide a valuable resource for academicians, digital forensics professionals, and researchers by enhancing their comprehension of the current state of the art in artifact recovery from social media applications. Additionally, it highlights the need for continued research to keep up with social media's constantly evolving nature and its consequent impact on digital forensics.

**Keywords:** artifact analysis; digital forensics; disk forensics; memory forensics; network forensics; social media forensics





## 1. Introduction

The term "Social Media" refers to a variety of interactive online platforms, chat rooms, and internet forums. They all have their own unique features and purposes that encourage seamless user connectivity, interactive information exchange, and data transfer via internet-mediated communications. Social media is becoming a vital aspect of modern civilization as a result of the broad adoption of new technology and the internet's pervasiveness in the lives of billions of people globally [1]. Some of the most popular social media applications include WhatsApp, Facebook, and Instagram. The COVID-19 outbreak and the resulting lockdowns further allowed deeper penetration of social media applications into users' daily lives. This made the growth of social media applications like TikTok even more prominent. Statistics for January 2023 state that 59% of the world's population uses social media for an average of 2 h and 31 min per day [2].

As a result of the extensive communication and widespread user engagement facilitated by social media applications, they have emerged as a new avenue for criminal activities known as social media-mediated crimes. These crimes are becoming advanced in nature, owing to the vast information exchange that takes place between millions of devices across the globe [3–6]. Social media applications give cybercriminals a platform to manipulate personal data and use it to perpetrate crimes [7]. Some of the crimes committed through social media platforms include spam (unwanted messages embedded with harmful links that lure users into giving personal information) [8], online identity theft (involves

taking someone's identity without their consent with the motive of committing fraud or financial theft) [9], cyberbullying (harassing, humiliating, or threatening another through the internet) [10], sexual exploitation (using someone's sexuality for personal or financial gain, often through coercion or manipulation), and many other crimes.

Digital forensics is the process of identifying, acquiring, processing, analyzing, and reporting on data stored electronically [11]. The combination of social media and digital forensics has given rise to a new field called Social Media Forensics (SMF) [12]. SMF is the process of collecting, analyzing, and preserving digital evidence from social media platforms. Over the last ten years, it has been acknowledged as a distinct branch of digital forensics. In legal cases concerning cyber crime where the perpetrator, victim, or witnesses may have used social media platforms, social media artifacts are essential as evidence [7,13]. Social media artifacts in the context of digital forensic investigations refer to the digital traces, remnants, or pieces of data left behind by using social media platforms. Common social media artifacts include chats, posts, geolocation, timestamps, deleted chats, and much more.

These artifacts can be valuable sources of evidence in various types of investigations. Trials involving the use of evidence from social media evidence are continuously increasing. In 2016, only in the United States, 14,000 decisions were observed, out of which 9500 heavily relied on evidence from social media [13], which is twice as high as the number in 2015. Due to the exclusion of cases in which social media content was used but no decision was made, it should be noted that these numbers are significantly lower than the actual number of investigations. They do, however, emphasize the undeniable significance of social media data.

One positive aspect of social media crimes is that criminals often leave digital footprints of their deeds, which is where social media forensics comes into play. Among various types of cybercrimes taking place, cybercrimes executed via social media platforms, also called online social network (OSN) crimes, have recently accelerated in number. Thus, there is a critical need for forensic analysis of digital platforms operating social media applications, as these platforms can be used for criminal activity, terrorism, and other unlawful actions. When properly explored for its potential, social media content can prove to be an outstanding source of digital evidence for digital forensics investigators. The information available about potential victims and suspects on social media is endless. It offers a dynamic dataset of user-generated information, such as posts, friend lists, images, geographical information, videos, demographics, and more.

In this article, we review the current state of research in social media forensics. It provides an overview of the current technical practices for the extraction and analysis of social media evidence. The primary objective is to identify the gaps in current practices and explicitly outline the future research objective for social media forensics. The rest of the paper is arranged as follows. Section 2 of this article will provide a brief overview and history of the domain of social media forensics, highlighting the importance of social media as evidence in legal proceedings. Section 3 highlights the parameters the study is structured upon. The methodology of the paper is explained in Section 4. Sections 5–7 will review research work on various analysis aspects such as memory, network, and disk, respectively. Section 8 examines emerging trends discerned throughout this review. Section 9 outlines some of the challenges faced in this domain and future research focus areas to address them. Finally, Section 10 presents our concluding remarks.

## 2. Background

The inception of using social media evidence was first reported in 2009 in the trial of the United States v. Drew in California. In this case, the convicted woman had allegedly created a fake MySpace (a social media application) profile, leading to the suicide of a young girl. However, the formal recognition of the potential role of social media evidence in litigation was brought to light by John G. Browning [14]. His research highlighted the increasing use of social media and scenarios where the utilization of evidence from

social media becomes an inherent aspect of legal proceedings. In 2011, Zainudin et al. [15] contributed to this growing field by presenting a comprehensive social media forensics investigation model, bolstering utilizing online social networking (OSN) data as evidence. Additionally, in 2014, Keyvanpour et al. [16] referred to social media forensics as digital forensics 2.0 and suggested that this sub-domain is the future of digital forensics.

SMF is a growing domain, having only existed for over ten years. A study conducted by Damshenas et al. in 2014 presented a review of emerging trends in digital forensics but refrained from listing social media forensics as one of the domains due to the scarcity of publications addressing this subject [17]. Additionally, [18] reviewed the situation of evidence acquisition, admissibility, and legal jurisdiction in the domain of social media forensics. Therefore, a review of the domain of social media forensics is needed to assess the current state of the field, which will help us examine the challenges faced and how we can address them [18]. The objective of this survey paper is to address the following research questions:

- What are the current state-of-the-art artifact recovery techniques used in digital forensics from social media applications?
- What are the trends in research related to artifact recovery in social media applications?
- What are the current gaps in the literature that need to be focused on?
- What are the future research directions for artifact recovery from social media applications in digital forensics, and how can these techniques be improved to serve the needs of digital forensics practitioners better?

## 3. Preliminary Information

In this section, we conduct a comprehensive analysis of different methodologies used by various studies in SMF. We examined these studies based on several research parameters to provide an in-depth analysis of the correlation of the applied methodologies to SMF artifact recovery, as shown in Figure 1. They are essential aspects that define and shape this research. These parameters include the research objective, the framework used to conduct the investigation, the analysis focus, the experimental setup, and the tools employed by the authors. They explain what the study is about, and how it was conducted. The parameters are further discussed in depth in the subsequent subsections.

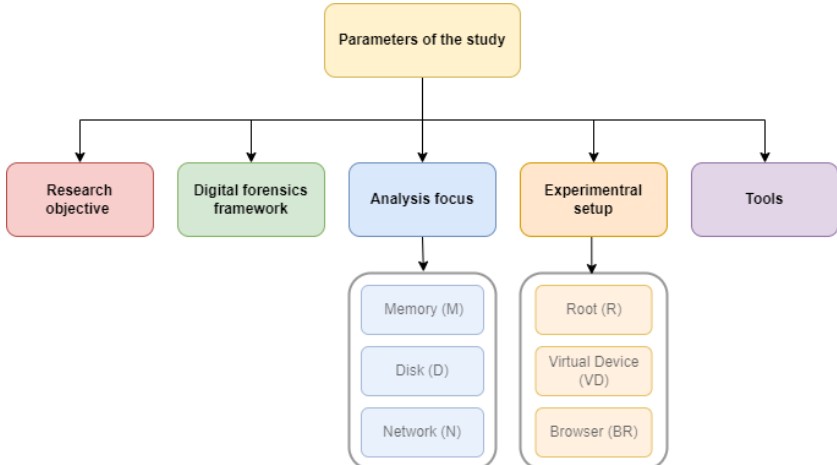

**Figure 1.** Study parameters.

### 3.1. Research Objectives

Numerous research initiatives within the field of social media forensics pursue various research objectives, from the recovery of digital evidence to the development of tools and the analysis of underlying databases and code. Below are the most common research objectives addressed, as shown in Table 1:

- **Artifact analysis**: Investigating digital traces and artifacts left behind by social media platforms upon conducting user activity.
- **Recovering deleted chats**: Recovering deleted chats on social media platforms to reconstruct digital interactions.
- **Decrypting messages/traffic**: Research on methodologies for decrypting encrypted messages and network traffic within social media applications.
- **Comparison of tools**: Evaluating different forensic tools on social media investigations, identifying their strengths and weaknesses.
- **Artifact correlation**: Establishing connections among different types of digital artifacts collected during the examination.
- **Tool creation**: Creating software for social media forensics.
- **Creating a forensic taxonomy**: Developing comprehensive taxonomies and categorizations to classify various types of digital evidence and artifacts encountered in social media investigations.
- **Database structure and analysis**: Analyzing the underlying structure of social media databases to gain insights into data storage and retrieval mechanisms.
- **Source code analysis**: Analyzing the source code of social media applications to uncover vulnerabilities, backdoors, or hidden features that may have forensic significance.

**Table 1.** Research objectives.

| Research Objective | References |
|---|---|
| Artifact Analysis | [7,19–125] |
| Recovering deleted chats | [126–128] |
| Decrypting databases/traffic | [40,72,83,129–136] |
| Comparison of tools | [47,51,56,109,112,127,128,137–141] |
| Artifact correlation | [67,77,134,142,143] |
| Tool creation | [33,93,144–149] |
| Creating a forensic taxonomy | [49,150,151] |
| Database structure and analysis | [36,52,131] |
| Source code analysis | [23,32,142] |

*3.2. Common Digital Forensics Frameworks*

Various digital forensic frameworks are employed to ensure methodical and structured investigations. These frameworks serve as invaluable resources for digital forensic practitioners, offering guidelines, protocols, and methodologies to ensure that investigations are carried out systematically in accordance with recognized industry standards. These frameworks are explained below:

- **National Institute of Standards and Technology (NIST)**: NIST offers a comprehensive framework that provides guidelines and standards for digital forensic investigations. It consists of four phases, namely collection, examination, analysis, and presentation [152], as shown in Figure 2.
- **Association of Chief Police Officers (ACPO)**: The ACPO framework is widely adopted in law enforcement agencies in the United Kingdom. It outlines procedures and best practices for handling digital evidence in criminal investigations [92].
- **McKemmish Framework**: Developed by Margaret McKemmish, this framework focuses on the digital preservation aspect of forensic investigations. It emphasizes the need to maintain the integrity and authenticity of digital evidence over time [153].
- **Digital Forensic Research Workshop (DFRWS)**: DFRWS is a community-driven organization that has contributed significantly to developing digital forensic standards and methodologies. Its framework consists of six stages, namely identification, preservation, collection, examination, analysis, and presentation [154].

- **National Institute of Justice (NIJ)**: The NIJ framework caters to the specific needs of the criminal justice community in the United States. It addresses forensic procedures, evidence handling, and the integration of digital evidence into the criminal justice system [155].
- **iPhone Forensic Framework (iFF)**: Existing commercial solutions and approaches in the field of iPhone forensics tend to be costly and complex, often demanding supplementary hardware for the investigative process. Consequently, Husain et al. [156] introduced a simple framework for iPhone forensic examination, comprising three main stages: data retrieval, data examination, and data presentation. This framework proved to be effective in extracting evidence from an iPhone.
- **International Digital Forensics Investigation Framework 2 (IDFIF 2)**: IDFIF 2 is an updated version of the IDFIF framework intended to enhance the global standardization of digital forensic practices. It focuses on promoting international cooperation and consistency in digital investigations [157].

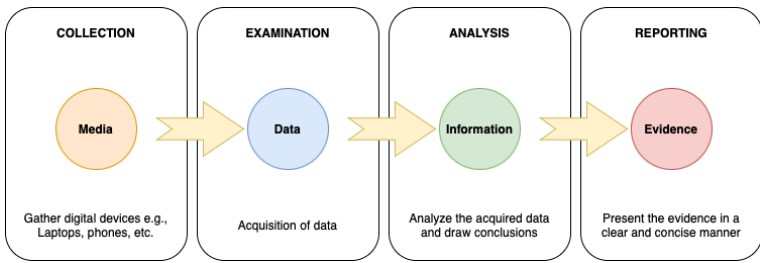

**Figure 2.** The NIST framework.

Among the most prevalent frameworks utilized by researchers are NIST [7,26,47,51, 52,68,84,95,97,98,102,103,109,117,124,127,138–140], ACPO [35,50,87], McKemmish [41,61], DFRWS [110,112,118], NIJ [56], iFF [21,156], IDFIF2 [90]. Out of all these frameworks, the NIST framework, as shown in Figure 2, is the most commonly utilized across the studies examined for this review. Its comprehensiveness, covering all aspects of digital forensics from evidence collection to reporting, makes it adaptable to diverse investigative scenarios. Additionally, NIST's commitment to regular updates ensures its relevance in an ever-evolving field. These factors collectively make the NIST forensic framework a preferred choice in the digital forensics community.

*3.3. Analysis Focus*

Social media artifacts can be found in various locations within a computing device. Researchers and digital forensics experts typically focus on three primary aspects when examining social media applications: the disk, memory, and network. Hence, we structure our review of existing literature to reflect these analysis aspects.

- *Disk*: The disk is essentially the storage of a device, primarily the hard drive and solid-state drives in computers and NAND flash chips in phones. The data in the disk provide numerous artifacts from social media applications, such as user-identifiable information, timestamps, media (photos and videos), chats, and much more.
- *Memory*: Memory refers to the volatile storage areas of a device, such as the Random Access Memory (RAM). Almost all applications use volatile memory to store data temporarily, such as the current state, open applications, active processes, etc. This provides access to real-time information, such as passwords, user activities, and more, making it valuable for investigations.
- *Network*: Analyzing network data involves monitoring and capturing network traffic exchanged. It allows investigators to track and analyze data in transit, potentially uncovering valuable evidence related to social media activities. This aspect is crucial as it involves real-time communication.

Figure 3 shows the frequency of the existing literature in each category of analysis focus surveyed in the paper. Additionally, the graph also depicts the frequency of each operating system under these categories. The figure depicts that the most tackled focus area of social media forensic investigations is the disk. The disk is commonly prioritized because social media applications store a substantial amount of user data on the disk, encompassing profiles, messages, posts, and multimedia content, even after users delete or modify their data. Moreover, unlike data in memory or network traffic, which are typically transient and may be overwritten or disappear once the device is turned off or the session ends, data on the disk are relatively stable. Additionally, comprehensive forensic tools and techniques are well-established for disk analysis, allowing for thorough analysis.

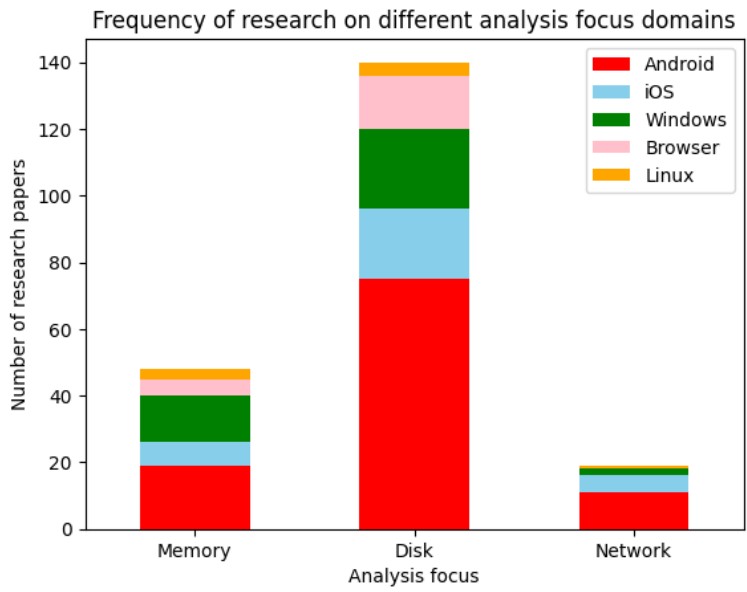

**Figure 3.** Frequency of surveyed literature based on the analysis focus.

### 3.4. Experimental Setup

The outcomes of various research experiments can vary significantly depending on the specific experimental setup chosen by the researcher. The common setup parameters employed by researchers in the field of social media forensics include:

- *Rooting or Jailbreaking*: One of the critical decisions researchers make is whether to root (for Android) or jailbreak (for iOS) the mobile device under investigation. Rooting or jailbreaking grants the researcher elevated privileges and access to parts of the device that are typically restricted. This decision can significantly impact the types of data that can be accessed and the methods employed for data extraction.
- *Virtual device environment*: Some experiments are conducted in a controlled environment using virtual devices or emulators. These virtual environments mimic the behavior of real devices and can be useful for testing and research without affecting physical devices.
- *Web browser*: Another approach involves conducting experiments through a web browser interface. This method can be advantageous for studying web-based applications and online social media user activities and the subsequent traces of evidence the browser leaves.

Figure 4 depicts the tools that are used for different experimental setups used by researchers. These tools enable the researchers to create the foundation for conducting their experiments. This figure is divided into three common experimental setups utilized: (a) conducting digital forensic analysis of browser data, (b) creating virtualized environments of a device (mobile device or desktop), and (c) rooting a mobile device. Under each category, we list the common tools utilized for each experimental setup.

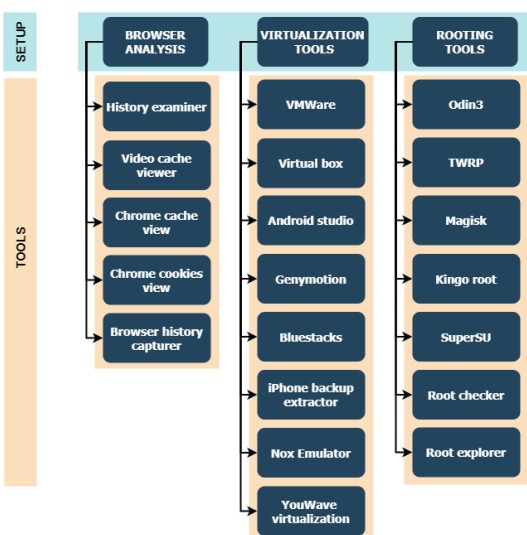

**Figure 4.** Common experimental setup tools.

## 3.5. Digital Forensics Tools

Social media forensics relies on a diverse array of specialized tools to analyze digital evidence effectively. These tools encompass a wide spectrum of functions for various digital devices. Each tool serves a unique role in the examination process, enabling researchers to dissect digital devices to reveal critical social media artifacts.

Figure 5 is organized by categorizing digital forensic tools based on the purpose of their utilization within the analysis focus areas (memory, network, and disk). The tools are further subdivided based on their specific usage within each focus area. For memory analysis, the tools are grouped into acquisition and analysis tools. In the case of network analysis, they are categorized as analysis and proxy tools, while for disk analysis, tools are further categorized into acquisition, analysis, and decryption tools.

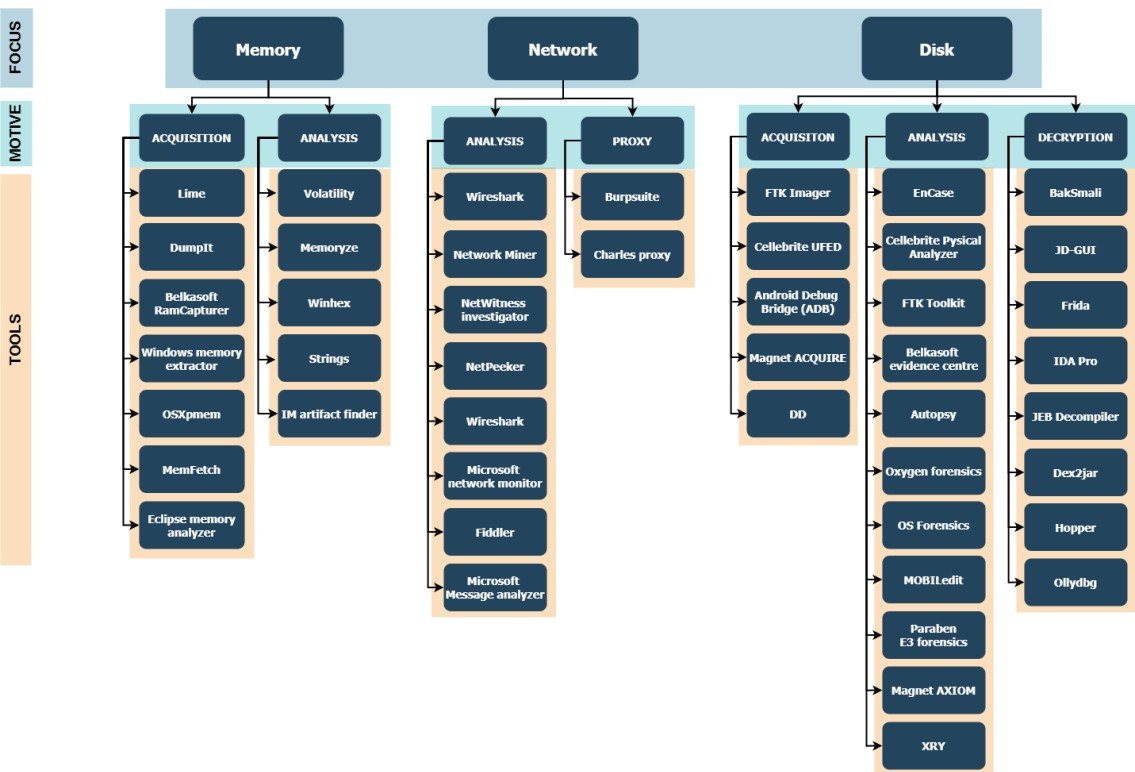

**Figure 5.** Common digital forensic tools.

## 4. Methodology

This paper presents a thorough review of approximately 170 research articles spanning the last 15 years, aiming to identify the literature on artifact recovery from social media platforms. The approach used in this study is illustrated in Figure 6, where "n" represents the number of articles. A search was conducted using electronic databases such as the Institute of Electrical and Electronics Engineers (IEEE) Xplore Digital Library, ACM Digital Library, Science Direct, and Springer Nature.

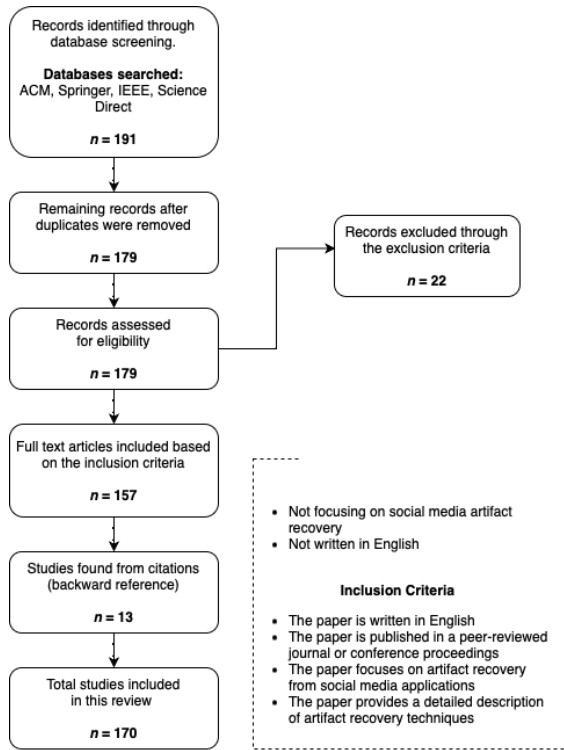

**Figure 6.** Methodology.

The search keywords included "artifact recovery", "social media applications", "digital forensics", "forensic investigation", and "social media forensics". The criteria for including papers were as follows: (1) the paper is written in English, (2) the paper is published in a peer-reviewed journal or conference proceedings, (3) the paper focuses on artifact recovery from social media applications in digital forensics, and (4) the paper provides a detailed description of the artifact recovery techniques and methods used. Subsequently, the chosen papers were evaluated based on the following criteria: (1) the type of social media application studied, (2) the techniques and methods used for artifact recovery, and (3) the contribution of the research in digital forensics.

*Organization of the Research*

In the following sections, we analyze existing the literature tackling artifact recovery on various social media applications, particularly following the structure outlined in Section 3.3. Hence, we have organized them to discuss the literature focusing on memory analysis in Section 4, while Section 5 discusses existing works that relate to network analysis, and finally, we thoroughly examine papers that perform disk analysis in Section 6.

Additionally, throughout the paper, we grouped the recovered artifacts into five distinct categories, as shown in Figure 7. Dividing the artifacts into separate categories helped organize evidence and findings from various studies. It also helped us analyze and discuss related findings together. These categories are detailed below:

- *User information*: This category contains artifacts that reveal critical data points on a user's personal information.
- *User activities*: This group of artifacts reveals information about user activities on social media platforms.
- *Metadata*: Metadata consists of crucial information like timestamps and geolocation, providing valuable context to other artifacts retrieved.
- *Password*: This category refers to the user account password being recovered.
- *Encryption key*: Encryption key artifacts are commonly recovered from studies focusing on database decryption of social media applications. They are used to decrypt the database.

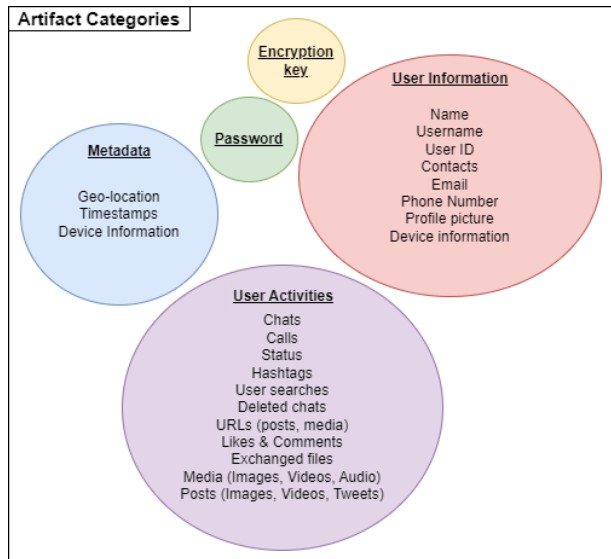

**Figure 7.** Artifact categories.

In each analysis focus category, we present a comprehensive overview of the reviewed literature through tables. These tables outline details such as social media applications studied, browser information (pertinent to web applications), utilization of a virtual device (VD), rooted status of the mobile device (R), tools utilized for acquisition and analysis, and the collected artifacts. To indicate the use of a virtual device or the rooted status of the device, we use "Y" for affirmative instances and "N" for negative instances. Furthermore, the indication of a specific browser and the recovered artifact categories is represented by a checkmark ("✓"), signifying usage, while a cross (×), denotes that the specified browser was not used, and the corresponding artifact category was not recovered. Additionally, we use "N/A" for certain information not provided in the original research reviewed.

The categories reviewed in this paper play a pivotal role in illuminating the methodologies employed in the studies under review. By meticulously outlining the parameters above, the review paper furnishes readers with a comprehensive understanding of the research methodologies. This detailed inclusion facilitates meaningful comparisons between diverse studies, enabling researchers to discern patterns, trends, and variations in methodologies. Such comparative analysis contributes to the synthesis of existing knowledge and aids in the identification of best practices within the realm of digital forensics. Furthermore, the incorporation of these categories serves as a diagnostic tool, allowing for the identification of gaps in the current body of research.

## 5. Memory Analysis Focus

Memory forensics is a branch of digital forensics that focuses on the analysis and extraction of digital evidence from a computer's volatile memory, also known as RAM. Volatile memory stores data temporarily while a computer is powered on and actively running [158]. Some of the data stored by the RAM include:

- program data (data related to currently running applications);
- process data (data related to currently running processes such as open files and data for execution);
- user data (data generated or modified by the users);
- network data (network connections);
- graphics data (video and graphics data including contents of the screen and graphics used in applications);
- user sessions (Information about user sessions, including user login credentials, active user profiles, and session-related data);
- browser data (data related to open tabs, history, cookies, and cached web content).

With a treasure trove of user and system information stored by the RAM, memory forensics is indispensable for investigating social media applications. Owing to its ability to capture a wide range of data, different researchers analyze the volatile memory of digital devices for various research purposes. The majority of the research is carried out to uncover what kinds of evidentiary artifacts related to social media applications can be found from the memory [57,59,60,76,78,159], whereas other researchers look for specific kinds of artifacts such as deleted chats [126] or encryption keys [83]. Additional research goals behind examining volatile memory for social media evidence are also to decrypt databases [129] and for the creation of tools for analysis of memory artifacts from social media applications [145,149].

*5.1. Memory Acquisition*

The memory forensics process typically involves two main phases: memory acquisition and memory analysis. Some of the most common tools used for memory acquisition in the literature include DumpIt [60,83,150] and LiMe (Linux Memory Extractor) [25,81,82,134], while other acquisition tools include FTK Imager [84], Android Debug Bridge [34,38,82], and Belkasoft Ram capturer [84]. From the review of the existing literature, it is seen that DumpIt is the most common choice for memory acquisition in Windows machines. It is a command-line memory tool that specializes in acquiring the contents of physical RAM primarily from Windows systems. The acquired memory (memory dump) is then output in a raw format, which can then be further analyzed using memory analysis tools. However, one of the tool's limitations is that it leaves a digital footprint on the memory [83], which can taint the memory dump acquired.

While DumpIt is the most prominent tool used for memory acquisition on Windows platforms, LiMe is the most prominent memory acquisition tool for Linux kernels and Linux-based devices such as Android. It is an open-source tool that can perform full memory captures. LiMe supports two memory acquisition methods, one via the transfer control protocol (TCP) network and the other via local storage, such as SD cards [81]. It is noteworthy that LiMe requires that the device be rooted to perform the acquisition [134]. This is because LiMe needs access to the kernel's memory space, which contains critical system information and data from running processes. Additionally, LiMe functions by loading a kernel module into the running kernel to create a memory snapshot. The access levels to perform all these functions are protected for security reasons. Thus, root access needs to be granted to capture memory using LiMe.

While some researchers prefer conducting experiments on physical devices, others use virtualization. Virtualization allows Windows systems to be configured on VMWare and Android Virtual Devices (AVDs) configured using platforms such as Android Mobile Device Emulator. When researchers use virtual devices, the process of acquiring a memory dump becomes streamlined. In the case of Windows systems, researchers can capture the memory by creating a snapshot, such as a .vmem file, while using VMWare, as performed by Chang et al. [59]. In the context of AVDs, researchers can bypass the need for device rooting since it can be preconfigured to grant root access to users within the virtual environment, as performed by Anglano et al. [134].

*5.2. Memory Analysis*

After acquiring a memory dump, memory analysis is the next phase. It is the process of examining the contents of the volatile memory to extract valuable information and evidence for investigative purposes. The most common tool for memory analysis is Volatility [33,81,83,85,126,134]. Volatility is a versatile open-source memory forensics tool. It provides a wide range of plugins to analyze memory dumps from various operating systems, such as Windows, Linux, and MacOS. Volatility can be used to extract information about running processes, network connections, registry keys, and much more. However, one of the major drawbacks of Volatility is the limited support for Linux and Mac operating systems. Analysis of these operating systems may require the researcher to create specific profiles for the particular operating system version in use.

Other than Volatility, many research methodologies prefer using hex editors to analyze memory dumps [25,37,59,60,81,83–85,145]. Hex editors are widely used for memory analysis for several important reasons. They provide a low-level representation of data, allowing investigators the opportunity to inspect the contents of the memory byte by byte. This level of granularity is required to identify data patterns needed to extract evidence. Another important reason for using hex editors is the ability to search for specific strings or patterns within the memory dump, which is one of the most employed methods used by researchers to look for evidence in the memory [59,77,78,81,83,84].

In the same line, the tool "Strings" is another popular tool for extracting a sequence of characters. A string of text is usually passed to search throughout the memory dump. The lines of the dump containing the matching text strings are then extracted. This is a traditional method used to analyze volatile memory [160]. Strings is commonly employed for this task as it supports large raw files, hexadecimal, ASCII, Unicode, and regular expressions. Other memory analysis tools used to conduct memory analysis in the literature include FTK toolkit [25,78,80] and EnCase [37,59].

In an effort to conduct a thorough examination of the remnants left by the LINE application on a Windows 10 system, Chang et al. [59] carry out investigations with different configurations of the environment. One of the configurations included conducting anti-forensic activities, such as deleting the application using CCleaner. This approach yielded a noteworthy discovery, revealing trace evidence of LINE activity, encompassing chats, usernames, and user files persisting in the system's RAM. Despite the relatively limited number of artifacts, the recoverability of artifacts remains intact.

While most of the memory analysis conducted on platforms is aimed at recovering evidence from social media applications locally downloaded on the device, some researchers have tackled memory forensics to recover evidence from browsers running social media web applications [25,58,94,97–101,148]. As seen in Table 2, the most targeted browser researchers use is Google Chrome because it is one of the most widely used web browsers globally, with a significant market share [161]. Its popularity makes it a prime target for forensic researchers because it represents a large portion of users' online activities. One of the most common research objectives related to browsers was to compare the artifacts uncovered from using social media web applications across different browsers [97–99]. The findings from these research experiments reveal that using different browsers can yield a discrepancy in recovered artifacts. This is due to variations in their architecture, data storage mechanisms, and how they manage user information. Hence, it is important to consider the browser's characteristics in any forensic investigation.

**Table 2.** Memory analysis on browser.

| Ref | Application | Browser | | | | VD | Tools | | Artifacts | | | |
| --- | --- | --- | --- | --- | --- | --- | --- | --- | --- | --- | --- | --- |
| | | Google Chrome | Firefox | Internet Explorer | Microsoft Edge | | Acquisition | Analysis | User Information | User activities | Metadata | Password |
| [25] | Facebook, Twitter, Google+, Telegram | × | ✓ | × | × | N | Lime | FTK Toolkit, HxD | ✓ | ✓ | ✓ | ✓ |
| [58] | Twitter | ✓ | × | ✓ | × | Y | N/A | Winhex, Memoryze, FTK Imager | ✓ | ✓ | × | ✓ |
| [94] | LinkedIn | ✓ | ✓ | × | ✓ | N | Mandiant | FTK Imager | ✓ | ✓ | × | ✓ |
| [98] | Instagram | ✓ | × | ✓ | × | Y | N/A | Winhex | ✓ | ✓ | × | × |
| [99] | Instagram | ✓ | ✓ | ✓ | × | Y | N/A | WinHex | ✓ | ✓ | × | × |
| [100] | TikTok | ✓ | × | × | × | N | DumpIt | HxD | × | ✓ | × | × |
| [101] | Google Meet | ✓ | ✓ | × | ✓ | Y | Volatility | Strings, FTK Imager | ✓ | ✓ | × | × |
| [148] | Facebook, Skype, Twitter, Hangouts, WhatsApp, Telegram | ✓ | ✓ | × | ✓ | Y | N/A | Strings, grep | ✓ | ✓ | ✓ | × |

*5.3. Artifact Recovery from Memory*

Most of the existing literature in the domain of memory analysis for social media evidence exists for the purpose of determining and exploring what artifacts can be uncovered upon analysis. We have illustrated the existing literature in Table 3. Upon surveying the literature, it is seen that many artifacts can be gathered from analyzing the memory. Some of these artifacts include chats [36,38,57,77,83–85,126,150], contacts [33,60,76, 77,149], media (URLs to photos, videos, images) [33,60,84,150], deleted chats [59], passwords [60,76,145,149,159], user profile information [83], geolocation data [84,150], and timestamps [59,77,126,150].

The chat feature is one of the most popular features in social media applications. It has become a central component of social media applications, contributing to user engagement. Chat features provide a convenient way to engage with other users in real time with options for multimedia sharing. Recovering chat artifacts is paramount in social media forensics due to the wealth of crucial evidence they contain. These chat records provide evidence of online interactions, offering invaluable insights into user behavior, relationships, intents, and activities on social media platforms. By examining chat artifacts, investigators can uncover evidence of cybercrimes, harassment, fraud, impersonation, and much more. Furthermore, these artifacts aid in verifying user identities and establishing a contextual understanding of events.

Passwords and encryption keys are crucial pieces of evidence that can be recovered from the forensic analysis of RAM (Random Access Memory). This is due to how computer systems handle sensitive data during their operation. When a user logs into a system or an application, their password or encryption key is temporarily loaded into RAM to facilitate authentication or data decryption. Even after the user logs out or the application is closed, fragments or residues of this sensitive information may persist in RAM for a certain duration. Modern operating systems and applications also use caching mechanisms to enhance performance, temporarily storing credentials in RAM. Moreover, when data are being actively used or processed, encryption keys must be loaded into RAM to decrypt those data on the fly, making them potentially accessible through RAM analysis.

Passwords hold the key to unlocking valuable evidence. They not only grant access to a user's social media profiles but also provide insights into their online activities, connections, and potentially illicit actions. In cases involving cybercrimes, cyberbullying, or online harassment, gaining access to a suspect's social media accounts can reveal critical evidence, including private messages, deleted content, and interactions with victims. This

information is indispensable for investigations, as it can help establish motives, uncover hidden activities, and facilitate the identification of culprits.

**Table 3.** Existing literature on artifact recovery from memory.

| Platform | Ref | Application | R | VD | Tools | | Artifacts | | | | |
|---|---|---|---|---|---|---|---|---|---|---|---|
| | | | | | Acquisition | Analysis | User Information | User activities | Metadata | Password | Encryption key |
| Windows | [57] | Digsby | N | N | N/A | Encase | ✓ | ✓ | ✓ | × | × |
| | [60] | LinkedIn | N | N | DumpIt | WinHex | ✓ | ✓ | × | ✓ | × |
| | [76] | Skype | N | Y | N/A | RSA keyfinder, AES Keyfinder, Volatility, Hex editor | × | × | × | ✓ | ✓ |
| | [77] | Facebook | N | N | Helix | FTK Toolkit, HxD | ✓ | ✓ | ✓ | × | × |
| | [83] | Google Hangouts | N | N | DumpIt | Volatility, WinHex | ✓ | ✓ | × | × | ✓ |
| | [84] | Line | N | N | Ramcapturer, FTK Imager | WinHex | × | ✓ | ✓ | × | × |
| | [126] | IMO | N | N | Custom python script | Volatility, Windbg | ✓ | ✓ | ✓ | × | × |
| | [145] | Digsby | N | N | N/A | WinHex | ✓ | × | × | ✓ | × |
| | [149] | Telegram | N | Y | Windows memory extractor | IM artifact finder | × | ✓ | ✓ | ✓ | × |
| | [150] | Skype, WhatsApp, Viber, Facebook | N | N | DumpIt | Strings | ✓ | ✓ | ✓ | × | × |
| Android | [33] | WhatsApp | Y | Y | Memfetch | Volatility | × | ✓ | ✓ | × | × |
| | [34] | Skype | Y | Y | ADB, DDMS | Eclipse memory analyzer, grep | × | ✓ | × | × | × |
| | [35] | Wickr | Y | N | Android tool memory dump | Strings | ✓ | × | × | × | × |
| | [36] | Wickr, Telegram | N | N | Memory dump app | String, grep | ✓ | × | × | × | × |
| | [37] | Line | Y | Y | N/A | Winhex, EnCase | ✓ | ✓ | × | × | × |
| | [38] | KIK | N | Y | ADB | Grep, JHAT | × | ✓ | × | × | × |
| | [78] | Viber | N | N | Android SDK | FTK Toolkit | ✓ | × | × | × | × |
| | [80] | Skype, MSN | N | N | Android SDK | FTK Toolkit | ✓ | × | × | × | × |
| | [81] | WeChat | N | N | Lime | WinHex, Volatility | × | ✓ | × | × | × |
| | [82] | Facebook, Viber, WhatsApp | Y | N | Lime, ADB | Custom script | ✓ | ✓ | ✓ | × | × |
| | [130] | Private text messaging, Wickr | Y | N | N/A | N/A | × | × | × | ✓ | × |
| | [134] | ChatSecure | Y | Y | Lime | Volatility | × | × | × | × | ✓ |
| Linux | [25] | Facebook, twitter, google+, telegram, openwapp, LINE | N | N | Lime | FTK Toolkit, HxD | ✓ | ✓ | × | ✓ | × |
| | [85] | Discord, Slack | N | Y | N/A | Volatility, WxHexeditor | ✓ | ✓ | × | × | × |
| iOS | [150] | Skype, WhatsApp, Viber, Facebook | N | N | DumpIt | Strings | ✓ | ✓ | ✓ | × | × |

## 6. Network Analysis Focus

The continually surging popularity of online services compels security experts and law enforcement agencies to seek innovative approaches for investigating cybercrimes and obtaining court-admissible evidence. There are a few researchers who have conducted forensic analysis on the disk in an effort to investigate encrypted databases of secured social media applications [125,132], but such approaches fall short when it comes to investigating end-to-end encrypted data. In such a case, network forensics comes in handy. Network forensics is a specialized branch of digital forensics that focuses on the collection, analysis, and interpretation of network traffic and data to uncover evidence related to cybercrimes and security incidents. It involves systematically examining network logs, packet captures, configuration files, and other network-related data sources to reconstruct events and recover network traffic artifacts [162,163]. Network traffic analysis is of paramount importance in the field of SMF. The existing literature solidifies this by showing that network traffic is a rich source of evidence for social media user interactions (posts, messages, and calls), as shown in Table 4. These data are crucial for reconstructing events, establishing timelines,

and identifying involved parties in forensic investigations by revealing insights into user behavior, connections, and engagement patterns.

**Table 4.** Existing literature on network forensics investigation for social media applications.

| Purpose | Ref | Application | System | R | Wireshark | Others | Artifacts |
|---|---|---|---|---|---|---|---|
| **Traffic characterization** | [43] | IMO | Android, iOS | Y | ✓ | N/A | Chats, Calls, Ports, IP add. |
| | [70] | Skype | Windows | N | ✓ | Netpeeker | Logins, Calls, Codec, Port |
| | [74] | Whatsapp | Android | N | ✓ | N/A | Chats |
| | [75] | Signal | Android | N | ✓ | N/A | Chats, Media, Calls, IP add. |
| **Traffic decryption** | [71] | Whatsapp | Android | N | ✓ | Pidgin | Calls, Phone no., Codec |
| **Artifacts** | [39] | Line | Android | Y | N | Logcat, Shark for root | Protocol, IP add. |
| | [40] | Telegram, Line, Kakaotalk | Android | Y | ✓ | Logcat | Timestamp, Protocol, IP add. |
| | [41] | Facebook, Twitter, Google+, Linkedin | Android, iOS | N | ✓ | N/A | IP add., Domain name, Timestamp, Protocol, Certificate |
| | [42] | Whatsapp, Viber, Instagram, Snapchat, Facebook | Android, iOS | N | ✓ | Network miner, Netwitness Investigator | Chats, Media, Location, Password, Server links |
| | [45] | Skype | Windows | Y | N | Microsoft message analyzer, Snooper | Calls, protocol, Codec, Phone no. |
| | [72] | Facebook, Twitter, Telegram | Firefox OS | N | ✓ | Network miner, Microsoft network monitor | IP add., Port, Certificate, Timestamps |
| | [73] | Telegram, Viber, Snapchat, Discord, etc. | iOS | N | ✓ | Charles proxy, Burp suite, Network miner | Chats, Location, Contacts, Password |

Social media applications facilitate the transfer of substantial data volumes across communication networks, encompassing various formats, with network packets being the most prevalent. Network packets hold useful user online activity data. When effectively captured, stored, and processed, they can yield valuable assets in forensic investigations and provide admissible evidence [164]. The de facto format for capturing network packets is libpcap. The Pcap Next-Generation Capture File Format (pcapng) has succeeded the traditional pcap format. The information extracted from these network packets can be used as evidence either directly or indirectly. For example, some information contained in the packets, including the sender and receiver IP addresses, port numbers, etc., along with the transferred data, can be used directly as evidence. In contrast, indirect information derived from multiple packets can also be used as evidence. This includes streams of packets sent from a particular host to another one in a certain pattern, which might indicate a specific user activity.

Many social media applications offer end-to-end encryption. These applications have attracted significant attention from users, driven by escalating concerns regarding their privacy. Notable social media applications, including Signal, WhatsApp, Facebook Messenger, and WeChat, have incorporated robust end-to-end encryption techniques during data transmission to safeguard user data's security and privacy. Signal, for instance, asserts the use of the highly secure Signal Protocol for communication. However, it is important to acknowledge that malicious actors also capitalize on the protective attributes of end-to-end encryption in these apps. Consequently, the presence of these security features presents an attractive medium for digital crime and fraudulent activities. Various researchers conduct network forensics for several reasons. Some of the most prominent reasons for conducting network forensics on social media applications include (1) traffic characterization [43,70,74,75], (2) traffic decryption [71], and lastly, (3) recovering artifacts [39–42,45,72,73], as shown in Table 4.

### 6.1. Common Research Aims for Network Forensics

Traffic characterization aims to identify user activities through the network traffic. The classification of user activities is performed by finding certain fixed patterns in network traffic. As most of the social media applications are secure and traffic flows are HTTPS-encapsulated, gaining access to the actual contents of information being exchanged between an app client and the servers is difficult. However, identification of a particular app and its user's activities is made possible by establishing behavior analysis of the traffic. This is performed by finding out a number of fixed patterns that are considered useful to identify the application over the network and to classify user activities.

The decryption of network traffic involves transforming encrypted data into their original, human-readable form. When data are transmitted over a network, they are often encrypted to protect their confidentiality and security. Decryption, therefore, serves as the means to unveil the content of these encrypted communications, making it comprehensible for analysis and investigation. To extract artifacts from network traffic, researchers establish a controlled network environment. Within this controlled setting, they simulate a sequence of user interactions within the application under examination. Subsequently, they capture the network traffic that results from these actions, meticulously dissecting and reconstructing evidentiary traces of potentially suspect data. This process allows for a comprehensive examination of digital footprints and potential forensic evidence within the network traffic, shedding light on user activities.

Many researchers have also incorporated the idea of using firewalls into the network forensic investigation [43,70,75,165]. Deploying a firewall within the investigation network enhances the ability to effectively monitor app behavior. Firewall rules are employed to verify the app's default behavior, enabling the imposition of restrictions and the identification of any hidden or alternative app behaviors. Additionally, this approach facilitates the observation of client–server connectivity design patterns, ports, and server ranges.

Using a firewall helps in understanding connectivity patterns by regulating traffic through different rule sets. A firewall can be used to restrict client traffic and compel the exposure of the client to alternate connectivity methods. Azab et al. [43,70] configured firewall rule sets to block out TCP ports that the application would regularly communicate on to understand the changes in network connectivity patterns. Moreover, firewalls can also be used to filter out traffic not concerning the experiment so that the researchers can focus on traffic corresponding to the experiment, as performed in [70,75]. Another use case of employing a firewall includes blocking server IP addresses, as performed in [75], which would result in reduced functionality of the application.

### 6.2. Common Network Forensics Tools

The heart of network packet analysis relies on packet capturing and analysis. One of the most utilized packet capture and analysis softwares used by researchers in the field is Wireshark [41–43,70,71,74]. In 1998, Gerald Combs introduced Ethereal, a packet analyzer that was later rebranded as Wireshark in 2006 [166]. Wireshark is a versatile open-source network protocol analyzer that can capture and analyze a vast array of protocols and traffic types. It can analyze protocols from simple HTTP/HTTPS protocols to complex protocols such as TCP, DNS, UDP, ICMP, etc. It has an exceedingly user-friendly graphical user interface (GUI) tailored for packet analysis [167]. This GUI features a packet browser capable of simultaneously displaying a list of packets, along with detailed information and packet bytes of the currently selected packet.

Other than Wireshark, Network Miner and Charles proxy are other common network packet analysis tools. NetworkMiner is a network analysis tool designed for passive network packet capturing and forensic analysis. Its primary function is to extract valuable information and artifacts from captured network traffic. NetworkMiner can dissect and analyze network packets to reveal insights such as IP addresses, domain names, usernames, file transfers, etc. It aids in reconstructing network conversations, allowing forensic analysts to piece together the chronology of network events. Additionally, The Charles Web Debug-

ging Proxy, developed by Karl von Randow in 2002, is a versatile web debugging proxy tool that primarily serves the function of monitoring and intercepting network traffic between a user's device and the internet. Its core purpose is to provide detailed insights into the HTTP and HTTPS traffic generated by web browsers or mobile applications. Charles Proxy allows users to inspect, analyze, and manipulate this traffic in real time.

*6.3. Network Forensics Artifacts*

Artifacts from network analysis primarily stem from monitoring and examining network traffic. These artifacts encompass data packets, communication logs, metadata detailing network interactions, and information related to IP addresses, ports, and protocols. IP addresses are fundamental to network forensic analysis. They help identify the source and destination of network traffic. However, we cannot solely depend on IP addresses for our investigation due to their dynamic nature. IP addresses often cannot be directly linked to a person [168] or a specific geolocation [169]. Some other prominent artifacts that can be gathered during a network forensic analysis include port numbers [43,72], protocols [39–41,45], domain names [41], certificates [72] used, and timestamps [39,41,72].

Port numbers help differentiate services and applications on a network, while protocols specify the rules and format of the network communication. They determine how data are structured, transmitted, and interpreted and help investigators understand the nature of network traffic. Certificates, specifically SSL/TLS certificates, are critical for securing web communications by encrypting data transmitted over HTTPS connections. They include details about the website's identity, encryption algorithms, and validity. Another common artifact retrieved is timestamps. They provide chronological information about network events.

Most of the authors of the existing literature focus on the artifacts from user activities related to chats [74] and calls [45,70,71]. This is because the most common user activities performed on social media applications are communication, such as chatting and calling. In light of this aim, Cents et al. [74] identified sent and received WhatsApp chat messages between a phone and the WhatsApp servers by detecting patterns in wiretap data. Wiretap data are utilized since it is difficult to trace any signs of network traffic monitoring by the suspect. Furthermore, Karpisek et al. [71] focused on decrypting WhatsApp network traffic to uncover information related to a call, while Azab et al. and Nicoletti et al. [45,70] examined the Skype application to characterize network traffic and retrieve artifacts related to calls, respectively. Some of the most prominent artifacts recovered from the above are audio codecs [45,70,71], call establishment and termination [70,71], call duration [71], and phone numbers [45,71].

## 7. Disk Analysis Focus

While conducting analysis on digital systems, it is paramount to analyze the storage media contained in these systems. This is because they are major sources of evidence, often holding a wealth of information that can provide critical insights into user activities. These storage media, including hard drives, solid-state drives, NAND chips (Android storage), and many more, serve as repositories of both active and historical data, making them central to the investigative process in digital forensics.

Disk analysis can be applied across a spectrum of platforms encompassing various operating systems, including Windows, Android, and iOS. Social media applications are often downloaded and installed on devices operating with these diverse systems. Each social media application possesses its own database, serving as the repository for user data, as shown in Figure 8. Researchers engage in digital forensic analysis of the disk to scrutinize these databases, conducting detailed investigations to recover artifacts and shed light on user activities within the social media sphere.

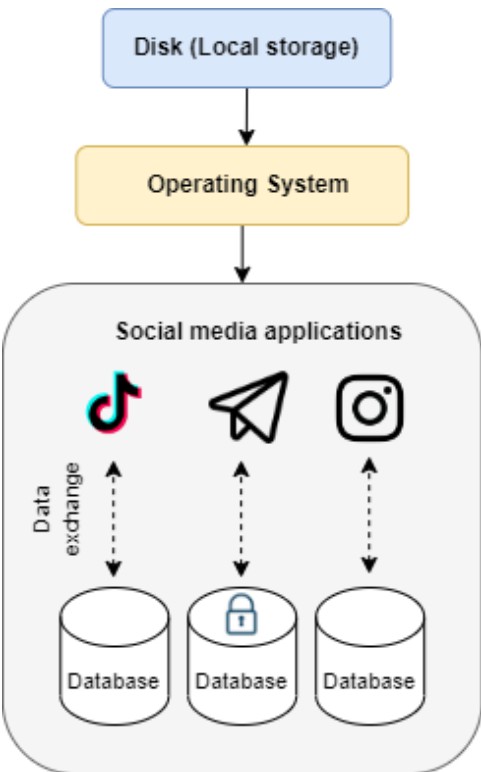

**Figure 8.** Key parameters in the digital forensic analysis of the disk: operating system and individual social media databases (sometimes encrypted).

*7.1. Experimental Setup*

The process of conducting disk analysis begins with the researcher preparing the platform they intend to use for experimentation. This platform can be either a physical computer or a virtual machine, depending on the nature of the investigation. On the other hand, in the case of mobile devices, researchers may also choose to root (for Android) or jailbreak (for iOS) the phones to gain administrative privileges and access to restricted areas of the device.

7.1.1. Virtualized Environments

To conduct digital forensics experiments, researchers either use physical devices or virtualized environments. Virtual environments rely on software that can simulate a physical device. In the rapidly changing landscape of technology, leveraging virtual machines for digital forensic analysis is increasingly advantageous. As technology evolves, new operating systems, file formats, and software environments emerge. Virtual machines adapt readily to these changes by enabling the creation of specialized, up-to-date analysis environments. This adaptability ensures that forensic analysts can keep pace with the latest technologies and forensic tools, ensuring their investigations remain effective and relevant in an ever-evolving digital world.

Virtualized devices, indeed, make it simple and cost-effective to run experiments on a variety of different virtual devices (featuring different hardware and software combinations). Furthermore, they allow a third party to use virtualized devices identical to those we used in our experiments, as well as to control their operational conditions so that the same conditions holding at the moment of our experiments can be replicated on them. In this way, repeatability is ensured.

Many researchers prefer to use virtual machines for several key reasons [30,46,50,53, 58,59,106,107,135,136,142,143]. Virtual machines provide a controlled and isolated environment for conducting forensic analysis. They also allow for creating a snapshot or clone to restore the original system or previous versions of the system, as performed by [30].

Additionally, they allow researchers to create replicas of the digital environment for use in different experiments [107]. Moreover, virtual machines can also be set up to accommodate various configurations for different experiments, as performed by [59].

VMWare Workstation is the most widely used tool for emulating desktop computers [59,61], whereas the most common virtualization tool researchers utilize to emulate an Android device is Genymotion [53,107,135]. Other utilized platforms include Android Studio [50], Android mobile device emulator [134,142], YouWave Virtualization [143], and Nox emulator.

Virtualization makes it easier to extract the data on the phone because virtualization tools implement internal storage as memory files, which can be acquired and analyzed. This makes the process greatly simplified, as it reduces the need for acquisition tools. For example, YouWave Virtualization implements the internal storage of the phone as a VirtualBox storage file [143] that can be parsed by a suitable tool for analysis. Additionally, virtualization platforms can be configured to grant root access to the virtualized devices [142], which removes the need for researchers to perform the tedious task of rooting the phone, hence making the experiment much simpler to conduct. Moreover, researchers also conduct virtualization in their experiment to look for any discrepancies in the results revealed from a physical device and a virtualized device, as performed in [135,143].

### 7.1.2. Rooting (Android)

Many researchers root their Android devices as part of the experimental setup before performing the experiments [39,40,105,106,111,127,128,140]. Rooting plays an important role in the forensic analysis of an Android device because it provides investigators with the elevated privileges necessary for accessing and retrieving data at the system level. Given that a significant portion of crucial files resides within the system partition of Android devices [120], rooting becomes a requisite step in obtaining vital evidence from the device. This step potentially allows for more data retrieval than if the device was unrooted [33]. Rooting not only allows access to protected directories containing user data (i.e., the/data/data directory), but it also allows users to back up some or all of the files located within these directories, making it easier for a logical acquisition to be conducted through backup applications. The most common tool used to root Android devices is Odin [7,66,123,124]. Other common rooting tools include Root explorer [37], TWRP (Team Win Recovery Project) [120,128], Root checker [122,140], Magisk [128,136], and Kingo root [115].

### 7.1.3. Jailbreaking (iPhones)

Just like rooting grants administrative access to Android devices, jailbreaking elevates iPhone privileges and removes software restrictions. Jailbreaking not only provides administrative privileges but also eliminates device software limitations. Ovens et al. [22,43] focused on analyzing the KIK messenger app to locate and examine artifacts created or altered by the application. To accomplish this, they conducted their investigation on an iPhone that had been jailbroken. By jailbreaking the device, they achieved the following objectives: (1) observed files modified or generated by the app during its operation, (2) bypassed the file system's access restrictions, and (3) installed and executed third-party applications essential for their analysis. In their research, the taig jailbreak tool facilitated access to the iOS file system, while Cydia, a software manager, allowed for the installation of necessary third-party tools not supported by Apple.

### 7.2. Disk Forensic Analysis Tools

In the realm of digital forensics investigations, researchers employ a diverse array of tools to extract and scrutinize social media evidence. These tools serve different purposes; some are dedicated to acquiring data from devices, while others are exclusively designed for data analysis. Nevertheless, many of these tools exhibit the versatility to execute both

acquisition and analysis tasks seamlessly. Moreover, it is worth noting that certain tools are tailored for specific operating systems, limiting their applicability to particular platforms.

Furthermore, within the spectrum of digital forensics tools, a distinction exists between those that are freely available or open-source and those that are proprietary, as shown in Table 5. This table is structured to provide a comprehensive overview by categorizing data according to the specific operating systems, the experiment's objectives, the retrieved artifacts, and the tools employed for both data acquisition and analysis. Furthermore, this table highlights the accessibility of these tools by subcategorizing them into two distinct groups, i.e., free and proprietary.

**Table 5.** Existing literature on disk forensics investigation for social media applications.

| OS | Purpose | Artifacts | Tools | |
|---|---|---|---|---|
| | | | **Acquisition** | **Analysis** |
| Windows | **Artifact Recovery** | User Activities [129–131] | **Free** | **Free** |
| | | | FTK Imager | HxD, WinHex |
| | | User Information and User Activities [26,32,60, 66,69,137] | **Backup** | **Proprietary** |
| | | | My Backup | Oxygen forensics, UFED PA |
| | | User Information, User Activities, and Metadata [27–29,59,93,147] | **Proprietary** | **Registry** |
| | | | Magnet AXIOM process, UFED Touch | Regshot, Reg decoder, Registry editor |
| | **Database Decryption** | [129–131] | Ollydbg, JEB compiler, IDA Pro, Hopper | HxD |
| iOS/Mac | **Artifact Recovery** | User Activities [64] | **Free** | **Free** |
| | | User Information and User Activities [21,23,24, 43,65–67,69] | iTunes | DB Browser for SQLite, pslist editor, HxD |
| | | User Information, User Activities, and Metadata [19,20,22,41,68,170] | **Proprietary** | **Proprietary** |
| | | | Cellebrite UFED, UFED Touch | Magnet AXIOM examine, UFED PA |
| | **Database Decryption** | [35,37,39,127] | Ollydbg, JEB compiler, IDA Pro, Hopper | HxD |
| Android (rooted) | **Artifact Recovery** | User Activities [35,37,39,127] | **Free** | **Free** |
| | | User Information, and User Activities [24,34,68,102,103,106,107,116–118,125,138,142] | ADB, My Backup Pro, Titanium backup, Helium backup | HxD, WinHex, DB Browser for SQLite |
| | | | **Proprietary** | **Proprietary** |
| | | User Information, User Activities, and Metadata [27,43,66,88,104,105,108–110,112,114,115,119,120,128,139,140,170] | Magnet Axiom Process, Magnet Acquire, MOBILedit forensic, UFED Touch, UFED 4PC, XRY | Oxygen forensics, Magnet Axiom examine, UFED PA |
| | **Database Decryption** | [39,134–136] | Ollydbg, JEB compiler, IDA Pro, Dex2jar | HxD |
| Android (non-rooted) | **Artifact Recovery** | User Activities [42,52,55,64] | **Proprietary** | **Free** |
| | | User Information and User Activities [41,48,53, 65,67,144] | Cellebrite UFED, Oxygen forensics, MOBILedit forensics, Magnet AXIOM process, Wondershare Dr.Fone, XRY | DB Browser for SQLite, SQLite Viewer, Autopsy, AccessData FTK, HxD |
| | | User Information, User Activities, and Metadata [36,47,49–51,54,56,143] | | **Proprietary** |
| | | | | Magnet AXIOM examine, Belkasoft evidence centre, UFED PA |

1. *Free tools:* Our analysis reveals that the choice of free data acquisition tools is contingent upon the operating system under examination. For Windows, FTK Imager emerges as the predominant option, while iOS investigations frequently employ iTunes, and Android device data acquisition commonly relies on ADB (Android Debug Bridge) [134] and backup utilities. Conversely, analysis tools exhibit a higher degree of consistency in their utilization across various operating systems. Hex editors and DB Browser for SQLite rank as the most widely used analysis tools, with a notable exception being plist editors, which are specifically tailored for examining

.plist files—these are key/value persistent storage files—found on iOS and macOS operating systems.

2. *Proprietary tools:* Proprietary tools represent closed-source software applications that are developed and exclusively owned by specific organizations. Typically, these tools necessitate the acquisition of licenses for authorized usage. Moreover, the outcomes produced by these tools are generally accepted in a court of law, making it difficult to dispute their findings. Notable players in the field of digital forensics software include Cellebrite, Magnet Forensics, Belkasoft, and Oxygen Forensics, among others. These companies often categorize their software offerings based on distinct functionalities. For instance, Cellebrite distinguishes between the Cellebrite UFED (Universal Forensic Extraction Device), tailored for data extraction, and the Cellebrite PA (Physical Analyzer), designed for in-depth analysis. Similarly, Magnet Forensics offers the Magnet AXIOM Process for data acquisition and the Magnet AXIOM Examine for comprehensive analysis [67,127,134]. Other proprietary tools renowned for their data extraction capabilities encompass XRY, MOBILedit Forensics, Wondershare Dr.Fone, and Belkasoft Evidence Centre.

The selection of tools varies depending on the research objectives. While the aforementioned tools are primarily employed for artifact recovery, a distinct set of software tools comes into play when the focus shifts to database decryption. This specialized category encompasses tools such as OllyDbg, JEB Compiler, IDA Pro, Hopper, and Dex2jar. OllyDbg is a debugger and reverse engineering tool. IDA Pro (Interactive Disassembler Professional) and Hopper are disassembly and reverse engineering tools. Lastly, Dex2jar is a set of tools and utilities used for Android application analysis and reverse engineering.

### 7.3. Disk Forensic Acquisition

Once the experimental setup is completed, researchers conduct their experiments by emulating user interactions on the social media application to elicit the application to generate and store data on the device's memory. The next step is to acquire an image (copy) of the device to preserve original evidence and recover relevant artifacts. To extract data from the device, researchers use one of the three acquisition methods: logical acquisition [19, 21,42,64,65,67,69,170,171], full file system [46,64], or physical acquisition [32].

1. *Logical acquisition*: Logical acquisition involves extracting data at a higher level of abstraction, which mainly includes specific files and data from the device. However, it does not capture deleted files or data stored in unallocated disk space. Logical acquisitions are commonly conducted using ADB and backup applications [36,39,40, 52,170,172]. These tools help researchers extract application-specific files, directories, and user data. Android Debug Bridge (ADB) is a command-line tool used for managing Android devices. ADB facilitates communication between a computer and an Android device over a USB connection or a network connection (Wi-Fi or Ethernet). Additionally, there are many backup applications that allow users to backup data— including application data—mainly to the device's internal memory, to an external SD card, or to some designated cloud storage. These data can then be analyzed using forensic tools.

2. *Full File system acquisition*: Full file system extraction is an acquisition in which all the data and metadata related to a device's file system are collected and preserved as part of an investigation. This method captures the complete hierarchical structure of files, directories, and associated file attributes, such as timestamps, permissions, and file sizes. On the other hand, physical acquisition involves the creation of a bit-for-bit copy or clone of the entire device, which yields more information than a logical extraction would [32].

3. *Physical acquisition*: A physical acquisition is a common type of acquisition conducted by researchers [64,111]. It typically provides more evidence than full file system acquisition [69] because it captures not only the file system structures but also the entire contents of the storage device at a lower level, including unallocated space, deleted

files, and fragmented data. Tools such as Cellebrite UFED are most prominently used for full file system and physical extractions [64].

### 7.4. Disk Forensic Analysis

Once the data are acquired, researchers then analyze the data to recover artifacts. The two main kinds of analysis procedures used by researchers are manual analysis [22,39,40, 69,143] and automated analysis.

1. *Manual analysis*: Manual analysis pertains to the investigator's non-automated (manual) efforts in searching for populated artifacts. Manual and automated digital forensics analyses differ in how they handle digital evidence. Manual analysis relies on human expertise, where forensic investigators actively examine evidence, search for relevant artifacts, and make informed judgments based on their experience. While this approach is flexible and customizable, it is time-consuming and requires specialized knowledge and skills. To conduct manual analysis, most researchers use DB Browser for SQLite to analyze the database files [111,115,118,128] and hex editors such as HxD or WinHex [27,28,59,60,111,147].

2. *Automated analysis*: Automated analysis relies on specialized software tools and scripts to process and analyze digital evidence without direct human intervention. Automated analysis is usually conducted by specialized tools such as Oxygen forensics, Cellebrite UFED Physical analyzer, and others [32,55,65]. Many research articles have employed proprietary tools for automatically analyzing social media data. The most commonly used tools for analysis are MOBILedit, Belkasoft Evidence Center, Oxygen Forensics, Cellebrite Physical Analyzer, Magnet AXIOM, and Internet Evidence Finder [56,109,112,123,128].

3. *Source code analysis*: While data analysis of social media applications is the most common way to retrieve artifacts in SMF investigations, Gregorio et al. [23,32] proposed a methodology that will supplement the analysis of artifacts with steps such as studying open knowledge sources (books, related blogs, technical papers) and the source code of the application. It is seen that this methodology yields a broad amount of information. Consequently, it becomes important to delve into open knowledge and dissect the source code to comprehend the data extracted from application artifacts. The collective implementation of these three steps streamlines analysis and traceability and also mitigates reliance on forensic tools. Although this analysis methodology yielded more artifacts than the artifact analysis step yielded alone, there are some limitations to this methodology. In some cases, it is not possible to apply some of the steps due to a lack of information in the open knowledge sources, information from non-trusted sources, or a lack of public source code.

### Windows-Specific Forensic Analysis

Here, we review the analysis of data and artifacts unique to the Windows operating environment. These review areas focus on the registry (a centralized database that stores configuration settings and options for both the operating system and installed applications) and Windows phone (a smartphone that runs on the Windows operating system). While Windows phones have been discontinued, it is noteworthy that certain researchers have undertaken digital forensic analyses of social media applications on the platform.

1. *Windows registry*: Analyzing the registry during a forensic investigation in Windows systems is crucial. The registry encapsulates a wealth of information that includes system configurations, user activities, and program execution records. Registry information can be extracted and examined from a forensic image, i.e., a disk copy of the original evidence. To that end, authors of [28,60] also analyze the registry during their forensic investigation. Some of the major tools used for registry analysis are Registry Editor and Regshot. Some of the artifacts revealed from registry analysis include information on the application, such as the model ID and install time [28,60]. Other prominent artifacts include contact photos retrieved from LinkedIn [60].

2.  *Windows Phone*: Besides Windows systems, researchers have also explored conducting forensic analysis on Windows Phones. While conducting a forensic analysis of WhatsApp data on a Windows phone, Shortall et al. [65] acquired data using the DD command. This was because, at the time of writing, no tool could be used to acquire data from a Windows Phone. A few years later, while analyzing Telegram on the same platform, Gregorio et al. [32] opted for a physical acquisition using Cellebrite UFED Touch. Both experiments involved analysis using the tools Cellebrite UFED Physical Analyzer and Oxygen forensics, but unfortunately, almost no artifacts were recovered. In the case of WhatsApp, the authors recovered media and an encrypted database, but for Telegram, no artifacts were recovered.

### 7.5. Aims of Disk Forensic Analysis

As earlier stated in Section 3.1, there are a range of research objectives fulfilled by conducting the analysis of the disk, such as artifact recovery, decryption of databases, reconstruction of chats, and creation of tools, among many more. Therefore, in this section, we elucidate the diverse objectives that motivate researchers to engage in disk forensic analysis.

### 7.5.1. Organization of Data

Social media applications store vast amounts of user-generated data. Thus, it is key to understand the folder structure to identify where these data are stored within the device, thereby making it easier to locate and retrieve relevant evidence. Azhar et al. [36] and Tri et al. [52] conduct digital forensics analysis on social media applications to understand the organization of data in social media applications.

Azhar et al. [36] analyzed the data structures of Wickr and Telegram. The authors selected these two applications due to their ephemeral messaging features. Various forensic analysis techniques were employed to retrieve artifacts from these applications. This is because Wickr and Telegram employ several security measures, and the nature of recovered artifacts differs based on the type of acquisition. The analysis phase consisted of analyzing the application file (.apk) and the data directories of the application. In Wickr, the authors extracted the "classes.dex" file from the application. The .dex file revealed all the class definitions used by Wickr, giving further insights into the operation of the application, such as the encryption mechanism used [35]. At the same time, the analysis of the data directory revealed the workings of the ephemeral function of Wickr, bringing light to the fact that Wickr stores its received messages in encrypted ".wic" files [35,36]. Additionally, the analysis of Telegram revealed the storage mechanisms of its normal and secret chats.

In the same vein, Tri et al. [52] conducted a forensic analysis to determine the structure of folders in the IMO application. A logical acquisition was conducted, which was manually analyzed, revealing the folder structure of IMO. The results revealed that the IMO data directory consists of six folders, out of which two have subfolders. These subfolders consisted of images and videos populated by user activities, which can be further analyzed to recover artifacts.

### 7.5.2. Artifact Analysis

A common purpose for conducting social media forensic investigations on the disk is to recover artifacts. Digital forensics artifacts are pieces of information that are left behind on digital devices as a result of user activities. Social media platforms generate a wide range of artifacts, such as chats, private messages, posts, comments, calls, and many more. These artifacts are essential components of digital forensic investigations and provide valuable evidence that can be used to reconstruct events, analyze user behavior, and establish a timeline of digital activities. Many researchers aim to retrieve artifacts when conducting digital forensic analysis of social media applications [26–32,60,65,66,69,137,147].

There is a plethora of artifacts that can be extracted from social media analysis by performing a digital forensics investigation on the disk. Additionally, due to the functionality of different social media applications, the extracted artifacts in an investigation

can differ. Thus, Azfar et al. [121] created a forensic taxonomy of thirty popular social media applications, classifying the extracted artifacts into four categories, namely User and contact information, Exchanged messages, Timestamps, and User location, as well as other artifacts.

### 7.5.3. Analysis of Privacy Features

Social media applications increasingly incorporate features like private chats and chat delete/unsend options to address privacy concerns. However, these features can also be misused for secretive or malicious purposes. For this reason, many researchers focus on analyzing artifacts related to private chats, unsent messages, and deleted chats [35,39,55, 111,127,128,173].

Satrya et al. [39] conducted an extensive analysis of both regular and private chat conversations within popular messaging applications such as Telegram, KakaoTalk, and Line on Android. In the case of Telegram, the researchers discovered that the contents of both regular and secret chats could be easily accessed and read using the SQLite Browser tool. For Line, the results revealed a complete absence of encryption for Line's regular chat artifacts. However, Line's hidden chat feature demonstrated a slightly different behavior. Although it also lacked encryption, it possessed a self-destruct mechanism that necessitated the timely acquisition of the chat data before they are irretrievably lost. In the context of KakaoTalk, the authors could only retrieve the last message within a chat activity.

Unsending chats is another privacy feature offered by social media applications. Hermawan et al. [55] analyzed an Android phone with the objective of identifying any retrievable artifacts associated with user actions involving the "unsending" of messages. The analysis was conducted on Skype, Viber, Snapchat, Facebook, Telegram, Line, Instagram, and Whatsapp using proprietary tools such as MOBILedit and UFED PA. The results reveal that artifacts of "unsend" messages can be found on all platforms except Line and Snapchat.

Many researchers have also researched the recoverable artifacts from deleted and disappearing chats. Vasilaras et al. studied the recovery of deleted chats on Telegram, Salamh et al. [127] examined the forensic artifacts of WhatsApp's "delete for everyone" feature, while Kumar et al. [111] analyzed the retrievable artifacts from Instagram's vanish feature (messages which would disappear). In all three research experiments, the recovery of artifacts was made possible by the presence of the Write Ahead Log (WAL). The WAL file serves as a form of journal, maintaining a comprehensive record of all transactions that have been executed but not yet applied to the primary database. Utilizing the WAL as a journaling mode ensures the integrity of the primary database by committing changes to a separate file until a checkpoint is reached. Examination of the WAL file yields insights into the most recent state of the database. For instance, data recently deleted and absent from the primary database may still be retrievable from the database's WAL file.

### 7.5.4. Reconstruction of Artifacts

Social media artifacts often contain critical evidence related to cybercrime and other malicious activities. However, simply collecting digital data is not enough. Understanding the context is vital. Decoding and interpreting the artifacts can help with the comprehension of the artifacts, providing insights into the chronology of the events that took place. Furthermore, the reconstruction of artifacts can prove to be invaluable in creating timelines, identifying patterns of behavior, and understanding the sequence of events. To this end, a few researchers have focused on reconstructing social media artifacts [40,134,143].

Anglano et al. [143] discussed the process of decoding and interpreting all the artifacts and data produced by WhatsApp Messenger on Android devices. They further illustrate how these artifacts can be correlated to deduce diverse forms of information that would remain incomprehensible if each were examined in isolation. Notably, the authors provide a detailed exploration of the structure of the contacts and chat databases, enabling the interpretation of stored data. These artifacts were subsequently correlated, revealing valuable insights, including the identification of added, blocked, and deleted contacts,

alongside the reconstruction of chat histories and their contents. This insightful analysis was rendered possible through the utilization of WhatsApp's log files, which record user activities and allow the authors to correlate information seamlessly between the log files and the databases.

In subsequent studies conducted by Anglano et al. [134] and Satrya et al. [40], the authors pursued a common objective, each focusing on distinct messaging applications—ChatSecure (an application that allows a user to communicate via multiple existing instant messaging accounts on a unified platform) and Telegram Messenger, respectively. Within their investigations, both research papers detailed the structural aspects of various application database tables, including components such as contacts, accounts, and messages. Consequently, these studies provide comprehensive guidance on how to analyze and correlate the data stored within the databases of ChatSecure and Telegram Messenger. This analytical approach yields insights into identifying IM accounts utilized by both the user and their friends in ChatSecure, as well as the reconstruction of messages, contacts, and file exchange chronologies specific to each application.

### 7.5.5. Decryption of Databases

Storing user information in databases on the disk raises significant privacy concerns, notably regarding the potential exposure of users' confidential data, including chats, photos, and personally identifiable information [174]. Typically, social media applications store user data within databases located on the device where the application is installed. One of the more common databases seen in this review is LevelDB, which is an open-source on-disk key-value store developed by Google. LevelDb is used by Microsoft Teams [31], Discord [29,30], Riot.im [30], and others. These database files contain extensive information about the user and their activities. To safeguard these data, social media applications commonly employ encryption measures, ensuring that only the application itself can access the stored information. Numerous researchers have investigated the decryption of these internal databases within social media applications [129,131,132].

Decryption procedures often entail deciphering the backup files associated with applications. These backup files are typically encrypted to protect user data. To retrieve artifacts, researchers utilize specialized techniques to decrypt these files. Once decrypted, investigators can analyze the artifacts, such as messages, images, and other user-generated content, shedding light on user activities.

Krishnapriya et al. [119] analyzed the Signal application data using an encrypted backup file to locate artifacts on an Android device. The authors manually acquired the backup file (.backup file) using ADB, which was then decrypted using a command line tool called Signal backup decryptor with a 30-digit passphrase. The database file in the backup was then analyzed for artifacts corresponding to user activity, such as user profiles, contacts, and messages.

Using a similar methodology of decrypting backup databases, Gudi et al. [133] decrypt WhatsApp backup databases (msgstore.db) on an Android device. The authors used a tool called "WhatsApp Key/DB Extractor" to extract the SQLite encrypted database in a decrypted format. The output is then processed by WhatsApp extract tool to output the database information in a human-readable format.

Additionally, Choi et al. [131] studied the backup process of KakaoTalk by reverse engineering the application to analyze the encryption process. They revealed that the key to the backup file could be generated using the user's password and a unique nine-digit number, which is assigned in the order of user registration on the app. If a weak password is used, the nine-digit number can easily be brute forced, leading to the encryption key [132]. Additionally, in a future study, Choi et al. [132] revealed that KakaoTalk and NateOn no longer required the user password to generate the encryption key. Instead, it required device-specific information such as the unique identifier, the model name, and the serial number. In the same line, Kim et al. [129] decrypted Telegram chat databases. It uses an

SQLite extension module called SQLCipher. The authors identified its parameters, which derived the encryption key.

Further analysis of research articles tackling the decryption of application databases revealed that the preferences.xml (and its variants) file could be a useful piece in the puzzle of decryption application databases [134–136]. Additionally, it is seen that many applications make use of SQLCipher to encrypt application databases [130,134,135].

Son et al. [136] research the decryption process of signal databases on a rooted Android device. Signal uses the Android Keystore to protect the encryption key. Thus, the authors developed an application to extract the key from the Android Keystore, as it cannot be extracted directly. To decrypt the Signal application's encrypted database ("signal.db"), you need to acquire a value called "pref_database_encrypted_secret" from the "org.thoughtcrime.securesms_preferences.xml" file. This value is a JSON string with the 'data' (cipher text and authentication tag), 'iv' (initialization vector) as a key, and its corresponding values, both of which are required for the decryption key. As a result, the database, multimedia, and log files could be decrypted.

Furthermore, Anglano et al. [134] focused on the methodology of decrypting Chat-Secure databases. ChatSecure employs SQLCipher for database encryption, with the encryption key being internally generated by the application, maintaining user confidentiality. This key remains securely stored within the device's volatile memory to facilitate decryption by ChatSecure as needed. However, to enhance security and prevent unauthorized access, ChatSecure utilizes the CacheWord library. This library encrypts the key using a user-defined secret passphrase and stores it within an XML file named "info.guardianproject.cacheword.prefs.xml", found in the "shared_prefs" directory. This layered security approach ensures that adversaries cannot decipher the databases using the saved secret key alone.

In a related context, Wu et al. [135] delved into the decryption procedures applied to WeChat's database. WeChat utilizes SQLCipher to encrypt chat message databases. Key data, such as chat records and configurations generated during WeChat's operation, are stored within three specific subdirectories, namely "databases", "shared_prefs", and "MicroMsg". The "databases" and "shared_prefs" directories house user authentication information and configuration files, while the "MicroMsg" directory stores crucial user activity data. When WeChat is initiated, it assigns a unique "uin" (User Identification Number) identifier to each user. Through analysis of the decompiled WeChat App code, the authors discerned that the decryption key is derived from the International Mobile Equipment Identity (IMEI) of the smartphone and the "uin" of the current WeChat user. Extraction of IMEI and "uin" data can be achieved from configuration files like "CompatibilityInfo.cfg" and "system_config_prefs.xml".

### 7.5.6. Creating Tools

Another application of conducting digital forensics analysis on social media applications is to create a tool. After thoroughly investigating the recovered artifacts and the structure of Discord's cache on various operating systems such as Windows, Linux, and Mac, Motylinkski et al. [147] developed a tool called DiscFor, which automatically retrieves all data stored on local Discord files, sparing the need for manual inspection of cache or JSON files. It can function both as a standalone Python script and as an executable file compatible with multiple systems.

Along the same lines, Anglano et al. [144] designed a software called "AnForA" that automates the activities carried out to forensically analyze Android applications. It begins by installing the target application onto a virtualized Android device. Subsequently, a series of experiments are conducted in which specific actions replicating the user interactions with the app are automatically executed within the application. The system then actively observes and monitors the device's file systems, allowing for the identification and correlation of data generated or altered during each action with the corresponding user interaction.

### 7.5.7. Browser Analysis

Performing forensic analysis on disk storage also enables the examination of social media web applications. Numerous researchers have engaged in forensic analysis of browser data stored on disks to recover valuable social media artifacts. The most common web browser chosen for such experiments is Google Chrome [86,90,96,175], likely attributed to its widespread usage, rendering it a primary focus for forensic investigations.

As depicted in Table 6, throughout the existing literature, the prevailing tool for acquiring web browser data has been FTK Imager, with an exception where iTunes is utilized to gather data from the Safari web browser [87]. As for the analysis phase, prominent tools encompass hex editors and DB Browser for SQLite, which are instrumental in delving into the database files of web browsers. In addition to these mainstream tools, specialized software such as ChromeCacheView, VideoCacheView, and Browser History capture tools have been developed with the specific purpose of analyzing web browser data. These tools are tailored to dissect various aspects of web browser data, including cache contents and browsing history.

**Table 6.** Disk analysis on browser.

| Ref | Application | Browser | | | | | VD | Tools | | Artifacts | | | |
| --- | --- | --- | --- | --- | --- | --- | --- | --- | --- | --- | --- | --- | --- |
| | | Google Chrome | Firefox | Internet Explorer | Microsoft Edge | Microsoft Edge | | Acquisition | Analysis | User Information | User activities | Metadata | Password |
| [86] | AIM, Meebo, E-buddy, Google Talk | × | × | ✓ | × | × | N | FTK Imager | FTK Toolkit | ✓ | ✓ | ✓ | × |
| [87] | AIM, Yahoo, Google Talk | × | × | × | × | ✓ | N | iTunes | DB Browser for SQLite, MobileSyncBrowser | ✓ | × | ✓ | ✓ |
| [88] | Facebook | ✓ | ✓ | ✓ | × | × | N | Encase | Encase | ✓ | ✓ | ✓ | × |
| [89] | Facebook | × | × | ✓ | × | × | Y | N/A | Internet Evidence Finder | × | ✓ | ✓ | × |
| [90] | WhatsApp | ✓ | × | × | × | × | N | FTK Imager | DB Browser for SQLite | ✓ | × | ✓ | × |
| [91] | Facebook | ✓ | ✓ | ✓ | × | × | N | N/A | FTK Toolkit | ✓ | ✓ | × | × |
| [93] | WhatsApp | ✓ | × | × | × | × | N | N/A | BrowSwEx | × | ✓ | ✓ | × |
| [94] | LinkedIn | ✓ | ✓ | × | ✓ | × | N | FTK Imager | FTK Imager | ✓ | ✓ | × | × |
| [95] | TikTok | ✓ | × | × | × | × | N | FTK Imager | FTK Imager, VideoCacheView, Browser History Capture | ✓ | ✓ | × | × |
| [96] | Discord | ✓ | × | × | × | × | N | N/A | DB Browser for SQLite, ChromeCacheView, HxD | ✓ | ✓ | ✓ | × |
| [98] | Instagram | ✓ | × | ✓ | × | × | Y | N/A | DB Browser for SQLite, WinHex | ✓ | ✓ | ✓ | × |
| [99] | Instagram | ✓ | ✓ | ✓ | × | × | Y | N/A | DB Browser for SQLite, WinHex | ✓ | ✓ | × | × |
| [100] | TikTok | ✓ | × | × | × | × | N | N/A | DB Browser for SQLite, History examiner, HxD, VideoCacheViewer | ✓ | ✓ | × | × |
| [101] | Google Meet | ✓ | ✓ | × | ✓ | × | Y | FTK Imager | ChromeCacheView, ChromeCookiesView, DB Browser for SQLite, Autopsy | ✓ | ✓ | ✓ | × |
| [176] | Youtube, Facebook | ✓ | × | × | × | × | N | N/A | ChromeCacheView, X-ways | × | ✓ | × | × |

## 8. Trends in Social Media Forensics

Social media platforms can capture and store massive amounts of user-generated content, making them valuable evidence sources in both civil and criminal cases. The reliability and timeliness of this evidence can play a significant role in determining the end result of a case.

The literature review findings reveal that Facebook [56,89,91], Whatsapp [33,65,143], and Skype [45,76,137] are the most commonly studied social media applications. These applications have been extensively studied due to their widespread use and popularity. This is also evident in the tools that are exclusively designed to analyze data from these specific applications, such as SkypeAlyzer [137] and Whatsapp Viewer [133], among others. Other popular social media applications, such as Instagram [53,110,111], Twitter [97,140,146], Signal [75,117,118], and Telegram [23,142,149], have also been extensively studied. It is worth noting that in some applications, such as Snapchat, very little evidence is recovered due to the ephemeral nature of the communication and the deletion of the artifacts [55,67].

Most research on social media applications has focused on the Android platform in mobile phones [54,55,124], as shown in Figure 3. This is because Android is the market dominator in the global smartphone market [121]. This makes it the prime target for malicious actors to conduct their activities, thus making it the most prominent environment for digital forensic scrutiny. Additionally, Android's open-source nature grants forensic experts deep insight into its underlying code, facilitating the development of specialized tools and techniques for analysis. The diverse hardware ecosystem on which Android operates presents both challenges and opportunities, demanding adaptability in forensic approaches to accommodate various device configurations. The platform's rich app ecosystem, user customization options, cloud integration, and evolving security features all contribute to its significance in digital forensics. Furthermore, legal considerations often place Android devices at the center of criminal cases, necessitating the continuous development of expertise in Android device analysis. However, with the increasing popularity of iOS devices, more attention is being given to the iOS platform [42,68,73]. It is also interesting to note that in mobile platforms such as Blackberry, no traces of evidence could be recovered, making it a very secure platform [7,66]. Although Blackberry platforms are the most secure, their usage has diminished over the years.

Regarding desktop computers and laptops, Windows is the most extensively studied platform [28,132,150], leaving a gap in the research on the Mac OS and Linux operating systems. The Windows operating system has earned its status as one of the most researched platforms in digital forensics investigations for several compelling reasons. Firstly, Windows has long been the dominant operating system for personal computers, meaning a substantial portion of digital data and potential evidence are hosted on Windows devices. Its ubiquity makes it a prime focus for forensic experts, as it is frequently involved in a wide spectrum of criminal activities. Additionally, Windows' complex and extensive architecture presents a rich landscape for forensic analysis, with various artifacts, logs, and registry entries offering valuable insights into user activities and system behavior. Furthermore, the wide range of Windows versions and configurations encountered in the field challenges investigators to continually adapt their techniques and tools, enhancing the need for ongoing research.

Examining social media applications on desktops or laptops additionally provides the opportunity to explore the web-based components of these platforms, offering a comprehensive view of users' online interactions and behaviors, both within the desktop application and across web interfaces. However, only a few studies have examined social media applications on web browsers [97,148,176], indicating a significant potential for further research in this area.

The analyses of memory, disk, and network are the three main focus areas encompassing the research analyzed in this survey. Among these, disk storage is the most frequently studied focus area [54], while network analysis has been the least studied [41]. Disk analysis is frequently studied in digital forensic investigations of social media applications due to the central role of local storage in storing user-generated data, making it a primary source of valuable evidence. Additionally, a wealth of specialized forensic tools exists to efficiently extract and examine data from storage devices, which have evolved to cater to the complex storage structures used by modern social media applications. On the other hand, network analysis is less commonly studied due to challenges such as the use of

encrypted communication, the volume of data, and their ephemeral nature. Network analysis is often secondary to disk analysis in typical investigations, where local storage holds the primary cache of user data.

Different analysis focus areas require specialized forensic tools for acquisition and analysis. For example, for memory analysis, the widely used tools for collection were Lime, FTK Imager, and DumpIt [82,97,150]. Volatility was widely adopted for analyzing memory [85]. The most commonly used tools to analyze network traffic were Wireshark and Network Miner [72].

Lastly, it was observed that although most research focused on recovering artifacts from social media applications, their purposes varied. While most articles concentrated on artifact recovery from common use cases of an application, such as online shop fraud [84], defamation crimes [95], pornography [56], cyberbullying [110], and web phishing [117], some papers specifically focused on artifacts related to particular application features, such as the "unsend message" feature [55], or a specific storage medium such as Indexed DB [93]. Furthermore, some articles also focused on creating a forensic taxonomy [49], decryption of databases [129,132], comparing tools [140], comparing different versions of applications [135], comparing platforms [66], artifacts recovered in different web browsers [97], detecting patterns in network traffic [74], creating tools [144,148], and reconstructing the sequence of chat messages [134].

Our findings in this review provide a comprehensive understanding of the various approaches and techniques used for artifact recovery. We also highlighted the need for continued research to improve the efficiency and accuracy of artifact recovery from social media applications in digital forensics.

## 9. Challenges and Future Research Focus in SMF

The increasing use of social media platforms has made artifact recovery a critical area of research in digital forensics. Our survey offers a comprehensive and systematic review of the current literature on artifact recovery from social media applications in digital forensics. The outcomes of this review were utilized to pinpoint crucial areas for further research in artifact recovery from social media applications that stem from the challenges identified.

1.  *Social media data in the cloud*: The field of social media forensics is developing quickly, and one aspect that has not been given much attention is the investigation of evidence stored in the cloud. With the increasing number of people using social media apps that keep their data in the cloud, it is now vital to concentrate on analyzing cloud data. However, cloud storage presents a significant difficulty for digital forensic investigators, as traditional forensic methods may not be enough to access and analyze cloud data [128]. Therefore, it is crucial to conduct research into the digital forensics of social media app cloud data to create more effective ways of recovering and analyzing artifacts. This research will enhance the efficiency of digital forensic investigations and help tackle the emerging challenges related to cloud-based digital evidence.

2.  *Lack of standard methodology for conducting social media forensics analysis*: It is crucial to create a comprehensive framework for social media forensics to guide future research [177]. While there are existing frameworks like NIST, NIJ, and ACPO that researchers use for digital forensic extraction and analysis, they are not tailored to the unique challenges presented by social media applications. Therefore, a new framework that specifically addresses the collection and analysis of data from social media platforms is necessary. This framework should offer a thorough approach to artifact recovery and tackle the unique challenges that arise from social media platforms.

3.  *Lack of specialized tools for social media forensics*: There is a need for further research on integrating social media data into traditional forensic tools. Most current digital forensics tools are not equipped to handle social media data effectively. Therefore, it is necessary to explore methods of integrating social media data into traditional forensics tools to enhance analysis and artifact recovery.

4.  *Vast amounts of social media data*: One particular area that could be addressed is the analysis of deleted and hidden data. Social media platforms allow users to delete or conceal their data, and it is essential to explore the potential for artifact recovery from such data. In addition, social media platform APIs can be used as a source of data for artifacts. These APIs offer a way to access the data stored on social media platforms, and their potential for artifact recovery in digital forensics has yet to be fully explored. Future research can focus on investigating these APIs and their potential for artifact recovery.

5.  *Heterogeneous and disparate sources of data*: They pose significant challenges for investigators and analysts. On social media applications, digital evidence is created in a variety of forms, including text, photographs, videos, and location-based information. Hence, the huge volume and disparity of data across many platforms makes it a difficult undertaking to efficiently acquire, analyze, and document this information. Investigators must deal with data consistency, dependability, and authenticity difficulties. Furthermore, individuals' differing privacy settings and data access rights hamper the recovery and investigation of digital evidence. As a result, dealing with the challenges of processing diverse and divergent data sources in social media forensics necessitates not just strong technological skills, but also a thorough awareness of legal and ethical aspects of the digital investigative process.

6.  *Adaptation of Machine Learning and Deep Learning models in SMF*: The use of machine learning models is highly promising for automating the process of artifact extraction from social media platforms. Specifically, deep learning models can be trained to identify relevant patterns and features within social media data, which can greatly enhance the efficiency and accuracy of artifact recovery. However, using these models may require technical expertise that some digital forensic professionals may not possess.

The examination of the current literature on artifact recovery from social media platforms in digital forensics emphasizes the importance of continued research. The future research directions outlined in this study can provide useful guidance for professionals and researchers working in digital forensics. Moreover, since social media applications are continuously evolving and introducing new features, research must be conducted to keep up with the rapidly changing landscape of social media forensics.

## 10. Conclusions

In this survey, we examined over 170 existing works in the literature tackling digital forensic analysis on several social media applications. We carry out an extensive examination delving into a wide range of analysis foci, research objectives, tools, and techniques relating to the field of social media forensics. We have structured this survey to emphasize that there are several research objectives behind conducting investigations on social media applications, such as artifact recovery, decryption of databases, and tool creation, to name a few. We also highlighted the most common digital forensic frameworks employed by most of the research reviewed. Subsequently, the reviewed papers were categorized into specific groups that outline the core research areas of the SMF investigation, particularly focusing on network, memory, and disk analysis. Furthermore, we delve into the platforms on which this research was conducted and the specialized tools subsequently employed for data acquisition and analysis. As a result, we developed a taxonomy for grouping the artifacts recovered during the investigation's analysis.

Our examination of existing research has illuminated prevalent trends in the field, simultaneously exposing gaps for future exploration. While data extraction from application databases on mobile devices has been extensively studied, a notable void exists in research addressing the retrieval and analysis of data from cloud storage—a prominent mode of data storage nowadays. Furthermore, our review underscores a lack of standardized methodologies or frameworks in the realm of digital forensics investigations of social media applications, where a conspicuous gap persists. Notably, the absence of a standardized methodology poses a significant challenge to the coherence of findings in this domain.

Lastly, despite the widespread manual techniques employed in analyzing social media data, there is a promising opportunity for future studies to leverage machine learning and deep learning models for the automation of large-scale social media data analysis. This potential shift towards automation could streamline and enhance the efficiency of digital forensics investigations in the ever-evolving domain of social media forensics.

**Author Contributions:** Conceptualization, K.G. and C.V.; methodology, K.G., and D.O.; validation, D.O.; formal analysis, K.G., D.O. and C.V.; investigation, K.G.; resources, C.V.; data curation, K.G. and D.O.; writing—original draft preparation, K.G. and D.O.; writing—review and editing, C.V., A.R. and N.S.; visualization, D.O.; supervision, C.V., A.R. and N.S.; project administration, C.V.; funding acquisition, C.V. All authors have read and agreed to the published version of the manuscript.

**Funding:** This research received no external funding.

**Data Availability Statement:** No new data were created or analyzed in this study. Data sharing is not applicable to this article.

**Conflicts of Interest:** The authors declare no conflict of interest.

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
