# Peer review of "A Comprehensive Survey on Artifact Recovery from Social Media Platforms: Approaches and Future Research Directions"

_information, doi:10.3390/info14120629_

Round 1

Reviewer 1 Report

Comments and Suggestions for Authors

In this paper, authors present a comprehensive survey over 170 reviewed articles related to the recovery of artifacts from social media platforms during the last 15 years, to understand the methodologies and social networks mostly studied in Social Media Forensics area. Likewise, they expose the gaps found in this area and that can be addressed in future works. I think this proposal is interesting, however, I consider it has some drawbacks that need to be addressed before a possible publication:

1. I strongly suggest that the authors write a brief explanation on the term "artifact" in the context of forensic social media, since it is not very understandable for the reader, especially when in Figure 7 presents categories of artifacts, but never explains what it is or what it refers to.

2. Concerning to the figures, I suggest improving their quality to have greater quality and understanding in the visualization part, as well as aligning the descriptive text of the figure.

3. In line 193, change 'exiting' to 'existing'

4. The blocks displayed in figures 4 and 5 do not clearly show the hierarchy and therefore it is difficult to understand the relation between what is written and what is shown in the image.

5. The methodology flow shown in Figure 6 is not clear and therefore not understandable, this figure and its description should be improved, for example, what does the parameter "n" mean, why these categories were selected and so on.

6. Table 2 is not previously referenced. In addition, why is there no information on the artifacts recovered from the second row? and why does not row 9 show the browser used? 

7. In Table 3, there are blank spaces in some columns, why? Authors should explain the reason behind the empty columns.

8. In addition, I find the description of the content shown in the tables used to gather the characteristics related to the memory, network, and disk sections to be poor.

9. Title description of Figure 8 does not match with the content shown in the image, you should change the description or content in the image.

Reviewer 2 Report

Comments and Suggestions for Authors

The paper is very well written, easy to follow, with good and appropriate illustrations throughout its figures and tables.

Its motivation is clear yet sufficiently well described, and overall the work it's able to summarize and analyze well a vast amount of related work.

The authors do a very good job at identifying their research objetives and at crossing them with proper literature as in Table 1.

Sections like 5 and 6 are well detailed and should be interesting for numerous audience.

Reviewer 3 Report

Comments and Suggestions for Authors

This paper reviews the current state of research in social media forensics. It provides an overview of the current technical practices for the extraction and analysis of social media evidence. The primary object is to identify the gaps in current practices and explicitly outline the future research objective for social media forensics. The idea is interesting. Howerver, there are still some problems in this paper that need to be modified in the following process.

(1) The paper is afflicted with formatting inconsistencies, and it is recommended that the author meticulously revise the manuscript's structure.

(2) The conclusion appears somewhat oversimplified. It is recommended to summarize the key points presented in the paper, ensuring that the future research plans or directions for further development are compelling.

(3)The references should be cited in a logical sequence to enhance the clarity and readability of the article.

(4)There are also some mistakes as followings in the manuscript, please check carefully.

(1) In Figure 1, it is appropriate to add some explanations of the parameters in the figure to make it easier to understand

(2) In Figure 3, it is suggested to represent the horizontal coordinates in line with normal reading habits.

(3) The format of tables in the manuscript is not uniform.

(5)It is suggested to be include more novel forensic algorithms in this paper. Such as:

[1]. Ma, B., Tao, Z., Ma, R., Wang, C., Li, J., & Li, X. (2023). A High-Performance Robust Reversible Data Hiding Algorithm Based on Polar Harmonic Fourier Moments. IEEE Transactions on Circuits and Systems for Video Technology.

Comments on the Quality of English Language

This paper reviews the current state of research in social media forensics. It provides an overview of the current technical practices for the extraction and analysis of social media evidence. The primary object is to identify the gaps in current practices and explicitly outline the future research objective for social media forensics. The idea is interesting. Howerver, there are still some problems in this paper that need to be modified in the following process.

(1) The paper is afflicted with formatting inconsistencies, and it is recommended that the author meticulously revise the manuscript's structure.

(2) The conclusion appears somewhat oversimplified. It is recommended to summarize the key points presented in the paper, ensuring that the future research plans or directions for further development are compelling.

(3)The references should be cited in a logical sequence to enhance the clarity and readability of the article.

(4)There are also some mistakes as followings in the manuscript, please check carefully.

(1) In Figure 1, it is appropriate to add some explanations of the parameters in the figure to make it easier to understand

(2) In Figure 3, it is suggested to represent the horizontal coordinates in line with normal reading habits.

(3) The format of tables in the manuscript is not uniform.

(5)It is suggested to be include more novel forensic algorithms in this paper. Such as:

[1]. Ma, B., Tao, Z., Ma, R., Wang, C., Li, J., & Li, X. (2023). A High-Performance Robust Reversible Data Hiding Algorithm Based on Polar Harmonic Fourier Moments. IEEE Transactions on Circuits and Systems for Video Technology.

Round 2

Reviewer 1 Report

Comments and Suggestions for Authors

I thank the author for considering the suggested comments to improve their proposal, however, there are still some drawbacks that I consider can be addressed and improved:

1. The text descriptions placed below the figures and above the tables are still aligned to the left, is this the format of the template or are the authors continuing to ignore this?

2. In the tables presented there are still empty columns, if no information from the original article is provided, I suggest writing an identifier or acronym (such as -, N/A or No data) and mentioning that it denotes that there is no information from the original.
